# Absolute Proteome Analysis of Hippocampus, Cortex and Cerebellum in Aged and Young Mice Reveals Changes in Energy Metabolism

**DOI:** 10.3390/ijms22126188

**Published:** 2021-06-08

**Authors:** Kinga Gostomska-Pampuch, Dominika Drulis-Fajdasz, Agnieszka Gizak, Jacek R. Wiśniewski, Dariusz Rakus

**Affiliations:** 1Biochemical Proteomics Group, Department of Proteomics and Signal Transduction, Max Planck Institute of Biochemistry, 82152 Martinsried, Germany; kinga.gostomska-pampuch@hirszfeld.pl; 2Department of Biochemistry and Immunochemistry, Wroclaw Medical University, 50-368 Wrocław, Poland; 3Department of Molecular Physiology and Neurobiology, University of Wroclaw, 50-335 Wroclaw, Poland; dominika.drulis-fajdasz@uwr.edu.pl (D.D.-F.); agnieszka.gizak@uwr.edu.pl (A.G.)

**Keywords:** glycogen, glycolysis, fatty acids, OXPHOS, total protein approach

## Abstract

Aging is associated with a general decline of cognitive functions, and it is widely accepted that this decline results from changes in the expression of proteins involved in regulation of synaptic plasticity. However, several lines of evidence have accumulated that suggest that the impaired function of the aged brain may be related to significant alterations in the energy metabolism. In the current study, we employed the label-free “Total protein approach” (TPA) method to focus on the similarities and differences in energy metabolism proteomes of young (1-month-old) and aged (22-month-old) murine brains. We quantified over 7000 proteins in each of the following three analyzed brain structures: the hippocampus, the cerebral cortex and the cerebellum. To the best of our knowledge, this is the most extensive quantitative proteomic description of energy metabolism pathways during the physiological aging of mice. The analysis demonstrates that aging does not significantly affect the abundance of total proteins in the studied brain structures, however, the levels of proteins constituting energy metabolism pathways differ significantly between young and aged mice.

## 1. Introduction

Aging affects several cognitive functions such as attention and memory, and it is presumed that it is an effect of disorders in the expression and proper localization of protein involved in synaptic transmission and plasticity [1,2,3]. However, a growing body of evidence suggest that age-related cognitive deficits may also result from the impaired energy metabolism of the brain [4].

It has been shown that in adult (12-month-old) mice, the capacity of the hippocampus to oxidize glucose in glycolysis and Tricarboxylic Acid Cycle (TCA) was elevated [4]. Aging has also been associated with an increased glycogen metabolism and oxidative phosphorylation (OXPHOS) and a reduced ability for free fatty acids utilization [4]. These changes were accompanied by the reorganization of the metabolism between cells: in the hippocampi of young animals, the high expression of glycolytic and glycogen metabolism enzymes and TCA proteins was attributed to astrocytes, while in adult mice, it was elevated in neurons. It has been thus suggested that in young hippocampi, astrocytes might deliver lactate to neurons to support their metabolism and/or synaptic plasticity formation but during aging, neurons become independent on astrocytic lactate and the metabolic crosstalk between the brain’s cells is disrupted.

The effects of aging on the level of proteins and gene expression in various brain structures of rodents have been investigated by microarrays [5,6] and mass spectrometry-based proteomic techniques [7,8]. However, except for Walther and Mann’s study [7], all the proteomic studies have been performed using semi-quantitative techniques. The study of Walther and Mann has revealed a relative stability of rats’ brain structures proteomes during aging [7]. More recent proteomic studies on age-associated changes in brain structures (e.g., [4,9]) are roughly in line with the results of Walther and Mann, however, they point to a significant alterations in the concentration of proteins of crucial energy metabolism pathways [4].

In the current study, we employed the label-free “Total protein approach” (TPA) method to focus on similarities and differences in energy metabolism proteomes in three structures (hippocampus, cerebral cortex and cerebellum) of young (1-month-old) and aged (22-month-old) murine brains. The role of aging-related physiological changes in the cerebral cortex and the hippocampus in cognitive performance is undisputed. However, it has been shown that changes within cerebellum also result in cognitive decline (e.g., see [10]). We quantified over 7000 proteins in each of the structures and, to the best of our knowledge, this is so far the most in-depth quantitative proteomic description of energy metabolism pathways during the physiological aging of mice. The results of the analysis demonstrate that aging does not significantly affect the abundance of total proteins in the studied brain structures, however, the levels of proteins constituting energy metabolism pathways differ significantly between young and aged mice. The physiological meaning of these changes is discussed.

## 2. Results and Discussion

### 2.1. Proteomic Analysis

The cerebellum, cortex and hippocampus were dissected from 1 month and 22-month-old mice (Figure 1A) and the tissues from each mouse were analyzed separately by global proteomics. The analysis of the spectra allowed for quantitation of 7547 proteins across the analyzed samples. Proteins that were identified with at least one unique peptide were used in the analysis. A principal component analysis of protein concentrations revealed age-related changes in the composition of proteomes in each of the analyzed tissue (Figure 1B). Notably, as expected, huge proteome differences were observed between the cerebellum, cortex and hippocampus.

### 2.2. The Effect of Aging on Total Protein Expression in Brain Structures

The expression data for 7324, 7088 and 7343 proteins in the hippocampus, cortex and cerebellum of old animals, respectively, were quantitatively determined and deposited to the ProteomeXchange Consortium via the PRIDE partner repository. For young animals, the data for 7347, 7323 and 7526 proteins in the hippocampus, cortex and cerebellum, respectively, were deposited therein (Figure 2A). Total molar protein concentration was statistically the highest in the cerebellum, in both the young and old animals (Figure 2B). The differences among the proteomes of individual animals in total protein concentration were negligible and were below 1%. Aging had no effect on the total protein concentration in the cerebellum (Figure 2B). On the other hand, in the hippocampus and the cortex of the old animals, the total concentration of proteins was significantly elevated (Figure 2B). However, the differences between the old and young animals were below 3.5% (Figure 2B).

The highest number of changes was observed in the hippocampus, where the concentrations of 2668 proteins were affected by aging (*p* < 0.05), while in the cortex and the cerebellum, there were 2228 and 2080 proteins, respectively, affected by aging (Figure 2A). Among these proteins, the titers of 165, 112 and 253 proteins in the hippocampus, cortex and cerebellum, respectively, of the old animals were increased at least twofold (Figure 2A). The number of proteins in which the concentrations in the aged animals decreased by at least two times were 691, 1072 and 569 in the hippocampus, cortex and cerebellum, respectively (Figure 2B). There were also 179, 242 and 175 proteins detected only in the hippocampus, cortex and cerebellum, respectively, of the old animals, and 401, 478 and 358 proteins that were present in the young animals’ hippocampus, cortex and cerebellum, respectively (Figure 2A).

### 2.3. The Fate of Glucose Molecules: Glycolysis, Glycogen Metabolism and Pentose Phosphate Pathway

Glycolysis is a basic and evolutionary primeval pathway in which glucose molecules may be oxidized under anaerobic conditions giving ATP and the reducing force NADH. Our quantitative analysis of glycolytic enzymes showed that their total concentration was significantly higher in hippocampus and cortex of old animals (Figure 3A). This titer was not altered by aging in the cerebellum, although, in this brain formation, the concentrations of regulatory glycolytic enzymes hexokinase (Hk1), phosphofructokinase (Pfkl) and pyruvate kinase (Pkm) and the main glucose transporter (Slca2a1) were significantly elevated in the old animals (Figure 3A). Hk1, Pfkl and Pkm were also increased in the hippocampi of old animals, while Slc2a1 and Pkm were elevated in the cortex of aged mice (Figure 3A). However, in the cortex, we also found that the amount of the main isoform of Pfk, Pfkp, was statistically significantly reduced.

In addition to the changes in the regulatory enzymes, we observed an increase in the titers of enzymes that are not supposed to be regulatory and/or rate limiting for glycolysis, such as enolase (Eno1 and Eno2), aldolase C (ALDOC) and glucose 6-phosphate isomerase (Gpi). The only significant decrease was associated with the cerebellar concentration of enolase 2 (Eno2) and phosphoglycerate mutase 2 (Pgma2). The titer of Pgam2 was, however, many times lower that the level of the main Pgam isoform, Pgam1, and thus, changes in its expression are not supposed to effect glycolysis and even the rate of the step the enzyme catalyzes in glycolysis.

The changes in glycolysis observed in brains of 22-month-old mice recapitulated those we have previously found in 1 year-old murine hippocampi [4]. The crucial changes were attributed to the expression of proteins which limit glycolytic flux: glucose transporter, Slc2a1 and hexokinase. Evidently, the capacity of the brain to oxidize glucose in glycolysis increases during aging.

### 2.4. Glycogen Metabolism

The activation of glycogen breakdown has been shown to play crucial role in the memory formation of young animals [11,12]. In turn, the inhibition of glycogen disruption was shown to improve the synaptic plasticity of old animals [13], and it was hypothesized that aging-related deficits in memory formation may result from an overexpression of glycogen phosphorylase (Pyg), the enzyme degrading glycogen into glucosyl units [4].

The present analysis showed that the aging-related increase in Pyg was not restricted to hippocampal formation but was also present in the cerebellum (Figure 3B). Both predominant Pyg isoforms, Pygb and Pygm, were significantly increased in the two structures. In contrast, the titer of cortical Pyg was not significantly altered by aging. However, also in cortex, the ability to form glycogen was elevated: the abundance of phosphoglucomutase (Pgm1), the enzyme redirecting glucose-6-phosphate into glycogen synthesis, was significantly elevated (Figure 3B).

### 2.5. Pentose Phosphate Pathway

The Pentose Phosphate Pathway (PPP) oxidizes the phosphorylated form of glucose, glucose-6-phosphate (G6P), to produce NADPH and ribosyl units for the synthesis of nucleotides. NADPH serves as a cofactor for the reduction of glutathione, which is the most abundant and efficient scavenger of hydrogen peroxide. The results presented here demonstrated that the capacity of the hippocampus and the cerebellum to synthesize NADPH was not affected by aging (Figure 3C). However, the cortical ability to produce NADPH was significantly reduced in the old mice. Interestingly, the ability to synthesize NADPH by the aged cerebellum was presumably even higher than that of the young mice because the concentration of the regulatory enzyme of the PPP, glucose 6-phosphate dehydrogenase (G6pdx), was significantly elevated by aging.

### 2.6. Pyruvate Fate: Lactate Synthesis and Pyruvate Dehydrogenase Complex

The final product of glycolysis, pyruvate, can be oxidized in mitochondria or reduced to lactate by lactate dehydrogenase (Ldh) and released from a cell through monocarboxylate transporters Slc17a3, also called Mct4. In this study, similarly as in [4], we did not detect the presence of Slc17a3 protein, however, we found two other members of the family of lactate transporters: Slc17a1 and Slc17a7 (Figure 4A). The transport of lactate from astrocytes to neurons has been shown to be indispensable for memory formation in the hippocampi of young animals [12]. Therefore, the lack of Slc17a3 is apparently in contradiction to literature data on memory formation. However, because all monocarboxylate transporters facilitate lactate transfer in both directions, thus, the other two monocarboxylate transporters may substitute Slc17a3 in its physiological function.

Overall, our analysis of lactate metabolism suggests the lack of significant changes during aging in the total protein expression (Figure 4A). However, we found that the level of Ldh A isoform (Ldha), which is suited to reduce pyruvate to lactate, was reduced in all the studied brain formations, although the changes were statistically significant only in the hippocampus (Figure 4A). On the other hand, the titer of Ldhb, the isoform that preferentially oxidizes lactate to pyruvate (and hence, it is presumed to participate in lactate uptake by cells), was increased in the cortex and the cerebellum (Figure 4A). In contrast to our previous study on the proteome of middle-aged mice [4], we did not observe a significant elevation of Ldhb in the hippocampus of old animals.

Similarly to our findings concerning middle-aged animals [4], we measured lower concentrations of practically all lactate transporters in the aged animals. However, the changes were statistically significant only for Slc17a1 in the hippocampus and Slc17a7 in the cortex (Figure 4A).

### 2.7. Pyruvate Dehydrogenase Complex

The pyruvate dehydrogenase complex decarboxylating pyruvate into acetyl-CoA connects glycolytic and oxidative glucose metabolism. In line with our previous study [4], we found that the total titer of proteins of the complex increases in hippocampal formation during aging (Figure 4B). A similar increase was also measured in the cerebellum, but not in the cortex (Figure 4B). The elevation of the pyruvate dehydrogenase complex components is not unexpected when we take into account the increased capacity of the hippocampus and cortex to produce pyruvate from glucose (Figure 3A) and from lactate by Ldhb (Figure 4A).

### 2.8. Glutamine/Glutamate Metabolism and Malate−Aspartate Shuttle

Glutamate is the main excitatory neurotransmitter in the brain, and it is formed from glutamine by glutaminase (Gls). The reverse reaction is carried out by glutamine synthetase (Glul) that catalyzes the ATP-dependent conversion of glutamate and ammonia to glutamine. Glutamate may also undergo decarboxylation by glutamate decarboxylase (Gad) that leads to the formation of γ-aminobutyric acid (GABA—the main inhibitory neurotransmitter in the brain), or may be reduced by glutamate dehydrogenase (Glud) to α-ketoglutarate, linking amino acids and the carbohydrate metabolism via the TCA cycle.

We found that concentration of glutamate dehydrogenase (Glud1) was higher in all the brain structures of the aged mice (Figure 4C), which indicates that glutamate and glutamine may be more important energetic substrates for the aged animal brain than for the young animal brain. Such an increase, however, was not correlated with the elevated abundance of Gls and mitochondrial glutamate transporters (Figure 4C).

Glutamate, except its role in neurotransmission and as a source of α-ketoglutarate for TCA, participates in the transport of reducing equivalents to mitochondria, forming the malate−aspartate shuttle (MA shuttle). The MA shuttle is the mechanisms of regeneration of NAD from NADH and the translocation of electrons produced in cytoplasm (e.g., in glycolysis) to mitochondria for oxidative phosphorylation. The MA shuttle is formed by two cytoplasmic enzymes, malate dehydrogenase 1 (Mdh1) and aspartate aminotransferase 1 (Got1), and their mitochondrial isoforms, Mdh2 and Got2. The transport of the shuttle, intermediated across the mitochondrial membrane, is facilitated by aspartate-glutamate exchangers (SLC25A12 and Slc25a13, also known as Aralar1 and Aralar2) and malate-α-ketoglutarate antiporter (Slc25a11).

In this study, we found that the concentrations of Got1 and Got2 were significantly elevated by aging in all brain structures (Figure 4C). This increase was accompanied by changes in the titer of Mdh2 in the hippocampus and the cortex, but its concentration was not statistically significantly elevated in the cerebellum. In contrast to Mdh2, the Mdh1 titter was not affected by aging (Figure 4C). The higher capacity to transport reducing equivalents from cytoplasm to mitochondria correlates well with our findings that glycolysis (a source of reducing equivalents in cytoplasm), was also elevated in the old brain structures (Figure 3A). Since the level of all antiporters involved in MA shuttle function (Figure 4C) were also elevated in all the structures of old brains (except cortical Slc25a11, in which the titer was unaffected by aging), it may be speculated that the transfer of glycolysis-derived reducing equivalents to mitochondria becomes more intense as the brain ages.

### 2.9. Fatty-Acid-Binding Proteins and β-Oxidation

In contrast to the carbohydrate catabolism proteome, the total concentrations of protein participating in fatty acids (FA) catabolism were either downregulated (Figure 5A) or unaffected by aging (Figure 5B). In all the studied brain structures, concentrations of proteins involved in the intracellular transport of long-chain fatty acids (fatty acid-binding proteins, such as Fabp3, Fabp5 and Fabp7) were significantly reduced by aging (Figure 5A). Such a decreased ability of the aged brain to carry FA in cytoplasm was, however, not reflected by a significantly lower concentration of β-oxidation enzymes (Figure 5B).

### 2.10. Tricarboxylic Acid Cycle and Oxidative Phosphorylation

The Tricarboxylic Acid Cycle (TCA, Krebs cycle) oxidizes acetyl-CoA and provides GTP and reducing equivalents (NADH and FADH), to produce ATP in the oxidative phosphorylation. Our study on brain aging revealed that the total concentration of proteins involved in the TCA was significantly elevated in hippocampus and cortex of old animals but not in cerebellum (Figure 6A). However, that increase was associated mainly with the elevation of the titer of only one protein—Mdh2, which also a part of the malate−aspartate shuttle that we found to be significantly increased in the brain structures of the old animals (Figure 4C).

In contrast to the hippocampus and the cortex, we found a significant elevation of mitochondrial citrate synthase (Cs; Figure 6A) in the aged cerebellum. This enzyme catalyzes the first step of TCA and its activity is regulated by the energetic state of a cell: Cs is inhibited by ATP, NADH and citrate and is activated by ADP.

Similarly, in the old animals, we observed increases in the concentrations of other enzymes regulating the rate of the TCA in ligand (ADP, Ca^2+^, ATP and NADH)-dependent manners, although their concentrations do not significantly affect the total amount of Krebs cycle proteins. We found that, in the hippocampus, a significantly higher level of expression was associated with oxoglutarate dehydrogenase L (Ogdh), isocitrate dehydrogenase subunit gamma 1 (Idh3g) and succinyl-CoA ligase subunit alpha (Suclg1) (Figure 6A). In the cortex, succinyl-CoA ligase subunit beta (Sucla2) was elevated while in the cerebellum, aging correlated with increases in Ogdh, Sucla2, Suclg1 and Suclg2 (Figure 6A).

Compared to 1-month-old mice, aging was associated with an elevation of the total proteins of oxidative phosphorylation (OXPHOS) in all the brain structures (Figure 6B). These changes were attributed to a significant increase in proteins of the ubiquinol-cytochrome-c reductase complex (complex III). As in the case of the 1-year-old hippocampus [4], we observed higher abundances of proteins of F1-F0 ATP synthase (complex V) than in the formation of the 22-month-old mice. We also found that the titer of all components of the complex V was significantly higher in the cerebellum of the 22-month-old mice than in the young ones, but in the cortex, the expression of the components of this complex was unaffected by aging (Figure 6B). Our analysis also revealed that the titer of cytochrome c oxidase complex (complex IV) was elevated in the cerebellum but not in the other brain structures (Figure 6B). Overall, the results of our analysis suggest the dysregulation of OXPHOS components expression rather than unidirectional changes.

### 2.11. Age-Related Changes in Concentration of Proteins Regulating Mitochondrial Dynamics

Energy production by mitochondria depends not only on the concentration of proteins of metabolic machinery (transporters of substrates, TCA, OXPHOS) but also on the dynamics of the mitochondrial network, i.e., on the processes of fusion/fission, biogenesis and motility.

Analysis of mitochondrial dynamics-related proteome in the tree brain structures did not reveal statistically significant differences in total amounts of the proteins between young and old mice. However, in old animals, we observed significantly lower titers of mitochondria motility-related proteins in the hippocampus, and fusion and biogenesis-related proteins in the cerebral cortex (Figure 7). Moreover, in these two brain structures, the total titers of proteins engaged in all the dynamics-associated processes were subtly but consistently lower in the old animals. The proteome of the cerebellum remained roughly stable. Only the titer of mitochondrial biogenesis-related proteins in the cerebellum of the old animals was about 15% higher than that of the young animals.

These observations contradict the data on absolute proteomes (see: “The effect of aging on total protein expression in brain structures” above) and might suggest that, although old animals are generally able to increase amounts of proteins (which, in part, might be a mechanism that compensates age-related defects in protein folding/quality check), their mitochondrial dynamics-related proteome is not so “adaptive”.

Among over 100 proteins comprising the mitochondrial dynamics proteome, statistically significant differences applied to 5 proteins in the hippocampus, 10 in the cerebral cortex and only 2 in the cerebellum (Appendix A). The titers of 16 of them were from 1.5 to 6 (usually over 2) times lower in old than young animals. Only one protein: Tfam (Transcription factor A, mitochondrial), showed a twofold increase in the cerebellum of old mice. Since the protein is required for mitochondrial DNA transcription and maintenance of proper level of mtDNA [14], its increasing titer, together with the increase of the total biogenesis-related proteome, might indicate efforts to ensure the adequate level of non-damaged mitochondrially-encoded proteins in the cerebellum of the old animals. A similar isolated rise of Tfam was observed earlier [4] in the biogenesis-related proteome in the hippocampus of the adult (1 year-old) mice.

In the hippocampus and the cerebral cortex of old mice, we observed lower titers of Nrf2, NSUN2, Fam120b, Yy1, MYCBP2, Crebbp and Akt2 (Appendix A). The transcription factor Nrf2 (2x lower titer in hippocampus of old animals) is involved in mitochondrial turnover by stimulating biogenesis of the organelles as well as the autophagy of mitochondria that failed to pass quality check. Nrf2 is also positively associated with mitochondria respiration and production of energy (for review see e.g., [15]). It is suggested that therapeutics that could activate the Nrf2 pathway may be effective in the treatment of neurodegenerative disorders.

Mammalian NOP2/Sun RNA Methyltransferase Family Member 2 (NSUN2; 2x lower titer in the hippocampus and cerebral cortex of old mice) belongs to the cytosine-5 RNA methyltransferase family that enhances sensitivity of the brain to oxidative stress and constitutes a link between stress an neurodevelopmental disorders in humans [16]. It is essential in mitochondrial tRNA methylation and ribosomal biogenesis, and defects in the nucleotide modification of mt-tRNA can lead to human disorders of mitochondrial respiration ([17] and citations therein). Apart from NSUN2, NSUN3 and 4 are also transported to mitochondria, and they are supposed to be more important than NSUN2 [17]. However, since only NSUN2 was identified in the hippocampus, maybe the importance of this family member should be reconsidered.

Fam120b (3× lower titer in the cortex of old mice) is also known as a constitutive coactivator of the peroxisome proliferation-activated receptor gamma (PPARγ). PPARγ regulates, among others, mitochondrial fusion–fission events, biogenesis and energy metabolism [18]. The E3 ubiquitin-protein ligase MYCBP2 (2× lower concentration in the cortex of old mice) is supposed to function as an activator of c-MYC [19], which in turn stimulates Tfam transcription. Crebbp (CREB binding protein; 6× lower titer in the cortex of old mice) enhances transcriptional activity of CREB toward cAMP-responsive genes, and cAMP signaling regulates, among others, mitochondrial dynamics (see: [20]). The activity of Akt kinases (~2× lower titer in the cortex of old animals) is necessary for mitochondria biogenesis and inhibition of the kinases exacerbates memory deficits [21].

Yy1 (Yin Yang 1; 2× reduced concentration in the cortex of old mice) is a transcription factor regulating mitochondrial gene expression and metabolism. It controls murine cerebral cortex development in a stage-dependent manner, and its importance decreases during development [22]. Thus, it might be concluded that the observed reduction of this particular protein titer during aging should not have any negative consequences for metabolism.

Among the proteins regulating mitochondria motility, the Kif2a, Wave-2 and Mark3 protein concentrations were significantly lower in the hippocampus, and Fez1, Map1s, Ttbk1 and Mark1 in the cortex of the old animals (Appendix A). Since there are numerous isoforms of these proteins, and their roles are often redundant, it is difficult to make unequivocal conclusions about the physiological meaning of the decreased titer of an individual protein for motility of mitochondria in old animals.

However, in some cases, the prominent changes applied to the most abundant isoform, or the whole protein group, and some of the proteins have specific functions that distinguish them from other isoforms. For example, Kif2a (a microtubule-depolymerizing kinesin) was the most abundant isoform of kinesin-like proteins in the hippocampus and had a ~23% lower titer in the old animals than in the young animals. MARKs 1–4 (microtubule affinity regulating kinases 1–4) are regulators of microtubule-dependent transport in axons. It has been shown that they phosphorylate microtubule-associated proteins that result in their detachment from microtubules and thus, remove obstacles from the microtubule tracks [23]. The titer of total MARKs was lower in the hippocampus and the cortex (~20% and over 50%, respectively) of old animals. High titers of these proteins in young animals, might facilitate the smooth transport of protein and vesicular cargo (including mitochondria) to neuronal protrusions. In turn, a microtubule-associated protein Map1s (2× lower titer in the cortex of old mice) is important in mitochondria quality control (causing irreversible aggregation of dysfunctional mitochondria and their mitophagy), and defects in Map1s function might lead to neurodegenerative diseases [24].

Previously [4], we did not observe statistically significant differences in the mitochondrial motility-related proteome between the young and adult (1 year-old) murine hippocampi, although we did find a significant decrease of some of the individual proteins with the increasing age. Here, in the hippocampi of old mice, the decrease of the motility-related proteome became statistically significant (Figure 7).

Taken together, in old animals, the reduction of titers of the proteins regulating mitochondrial network dynamics might point to reduced biogenesis/turnover and cellular transport (and thus, positioning in neuronal projections) of mitochondria, especially in the cerebral cortex (Figure 7). This may result in a decrease of their metabolic efficiency and lead to disturbance in the energy-demanding process of synaptic transmission, declines in cognitive functions and neurodegenerative diseases.

During aging, the cerebellum has the most stable mitochondrial proteome and the few observed changes are somewhat similar to those observed in the adult murine hippocampus [4].

In the other two structures, we did not observe such conspicuous changes as might be expected based on the previous results from adult murine hippocampi. This might suggest that only the mice with the most stable proteome were able to cross the barrier of adulthood and reach the old age.

Moreover, we did not find one common pattern of old age-related changes among the three examined brain structures. It might be thus concluded that each one of them has a different potential to defy aging.

### 2.12. Concluding Remarks

An increasing number of experimental results suggest that age-related cognitive impairment may result from disturbed energy metabolism of brain.

In this paper, we deliver the most in-depth quantitative description of hippocampal, cortical and cerebellar energy metabolism proteomes of young and old mice so far.

As would be expected, our analysis revealed significant differences in the proteomes among the three studied structures. Moreover, what is important is that it demonstrated that although the total protein expression in the brain structures is practically unaffected by aging, there are significant differences between young and old mice in the expression of numerous proteins involved in energy metabolism.

We found that that aging correlates with an increased capacity of all the studied brain structures to oxidize glucose in the glycolytic pathway, although this was expressed by changes in titers of slightly different set of regulatory proteins in each of the structures. Interestingly, that increase was not associated with changes in lactate metabolism proteins, but with elevation of pyruvate dehydrogenase complex concentration. This complex converts the product of the glycolytic oxidation of glucose, pyruvate, into acety-CoA, which serves as the main substrate for mitochondrial energy production pathways—TCA and OXPHOS. We observed that the increased ability to produce acetyl-CoA from glucose correlated with higher titers of TCA and OXPHOS proteins in the aged animals. However, in contrast to glycolytic pathway and TCA, changes in expression of OXPHOS complexes exhibited uncoordinated rather than unidirectional changes that suggest dysregulation of that pathway.

Glycolysis can stimulate oxidative metabolism not only by delivering pyruvate but also NADH, which may enter OXPHOS via the malate−aspartate shuttle-dependent transport of reducing equivalents to mitochondria. The results of our analysis demonstrated that the concentrations of most of the components of the malate–aspartate shuttle were significantly elevated in the old animals, which emphasized the increasing importance of glucose as the main substrate for ATP synthesis for aged brain.

In contrast, the concentration of fatty acids metabolism proteins was significantly reduced in old brains. The concentrations of both fatty acid binding proteins and β-oxidation enzymes was reduced by aging indicating the lower ability of old brains to utilize lipids as the energy substrate.

TCA and OXPHOS activities might be supported also by glutamate/glutamine-derived α-ketoglutarate. However, although we found that concentration of glutamate dehydrogenase was higher in aged brain structures, the titers of the enzyme converting glutamine to glutamate and proteins transporting glutamate to mitochondria were not altered by aging.

We also did not observe age-dependent changes in the capacity of the Pentose Phosphate Pathway. However, we found significant elevations of the titers of proteins involved in glycogen breakdown. This is in line with results of our previous study of the proteome of 1 year-old mice hippocampus, and corresponds well with the hypothesis that the alterations in glycogen metabolism may be a part of aging-associated rearrangement of astrocyte-to-neuron metabolic cross-talk.

Since we did not observe the existence of one common pattern (at the level of specific proteins) of the energy metabolism proteome changes in the three studied structures, it can be concluded that each of them has a unique potential to adapt to the general changes that occur in an organism during aging.

## 3. Materials and Methods

### 3.1. Animals and Tissue Preparation

The experiments were performed on two groups of female C57BL/10J mice: young (P30, *n* = 5) and aged (22 months, *n* = 5). Animals were treated as we described before [3], i.e., they were anesthetized with isoflurane and then decapitated. Brain formations were isolated in an ice-cold buffer containing (in mM): NaCl 87, KCl 2.5, NaH_2_PO_4_ 1.25, NaHCO_3_ 25, CaCl_2_ 0.5, MgSO_4_ 7, glucose 25 and sucrose 75; pH 7.4. The right hippocampi, right half of frontal cortex and right half of cerebellum from each animal were analyzed using quantitative proteomics.

### 3.2. Preparation of Tissue Lysates

Immediately after isolation, the brain structures from the right half of brain—the hippocampus (the whole right hippocampus), cerebellum (right cerebellar hemisphere) and cortex (the whole right isocortex)—were homogenized in the buffer containing 0.1 M Tris/HCl, 2% SDS, 50 mM DTT, pH 8.0 and incubated for 5 min at 99 °C. The samples were then cooled in liquid nitrogen and stored at −20 °C until proteomic analysis. Total protein was determined by measuring tryptophan fluorescence as described previously [25].

### 3.3. Multi-Enzyme Digestion Filter-Aided Sample Preparation (MED FASP)

Brain formations’ extracts containing 80 μg of total protein were processed using the MED FASP method [26] with modifications described recently [27]. Firstly, the proteins were cleaved overnight with LysC and then digested with trypsin for 3 h. The enzyme to protein ratio was 1:40. Digestions were carried out at 37 °C in 50 mM Tris-HCl with the addition of 1 mM DTT, pH 8.5. Aliquots containing 8 µg of total peptide were concentrated to a volume of ~5 µL and were stored frozen at −20 °C until mass spectrometric analysis.

### 3.4. Liquid Chromatography—Tandem Mass Spectrometry

Analysis of peptide mixtures was performed using a QExactive HF mass spectrometer (Thermo-Fisher Scientific, Palo Alto). Samples containing 8 μg of total peptide were separated on a 50 cm column with 75 µm inner diameter packed C18 material (100 Å pore size; Dr. Maisch GmbH, Ammerbuch-Entringen, Germany). The peptides were separated using the two-step acetonitrile gradient: 5–40% over the first 85 min (300 nL/min) and 40–95% for the following 15 min (300 nL/min). The temperature of the column was 55 °C. The mass spectrometer was operated in data-dependent mode with survey scans acquired at the resolution of 50,000 at *m*/*z* 400 (transient time 256 ms). Up to the top 15 most abundant isotope patterns with a charge ≥+2 from the survey scan (300–1650 *m*/*z*) were selected with an isolation window of 1.6 *m*/*z* and fragmented by HCD with normalized collision energies of 25. The maximum ion injection times for the survey scan and the MS/MS scans were 20 and 60 ms, respectively. The ion target value for MS1 and MS2 scan modes was set to 3 × 106 and 105, respectively. The dynamic exclusion was 25 s and 10 ppm. The mass spectrometry data has been deposited in the ProteomeXchange Consortium via the PRIDE partner repository [28] with the dataset identifier: PXD025978 (for the review: username: reviewer_pxd025978@ebi.ac.uk; password: QL3YI7nP).

### 3.5. Proteomic Data Analysis

The MS data was analyzed using MaxQuant [29] v1.2.6.20. Proteins were identified by searching MS and MS/MS data of peptides against the UniProtKB/Swiss-Prot database Carbamidomethylation was set as the fixed modification. The maximum false peptide and protein discovery rate was specified as 0.01. Protein abundances were calculated using the “total protein approach” (TPA) method [30,31]. The calculations were performed in Microsoft Excel using the relationship:(1)$$c\left(i\right)=\frac{MS-signal\left(i\right)}{Total\;MS-signal\times MW\left(i\right)}\left[\frac{\mathrm{mol}}{g\;\mathrm{total}\;\mathrm{protein}}\right]$$

### 3.6. Statistical Analysis

All results are presented as mean ± SEM unless we stated otherwise. The statistical analysis was performed using Student’s *t*-test preceded by Fisher *F*-test. We used non-paired Student’s *t*-test for comparisons between any two experimental groups. The statistical analyses were performed using SigmaPlot 11 software (Systat Software).

## Figures and Tables

**Figure 1 ijms-22-06188-f001:**
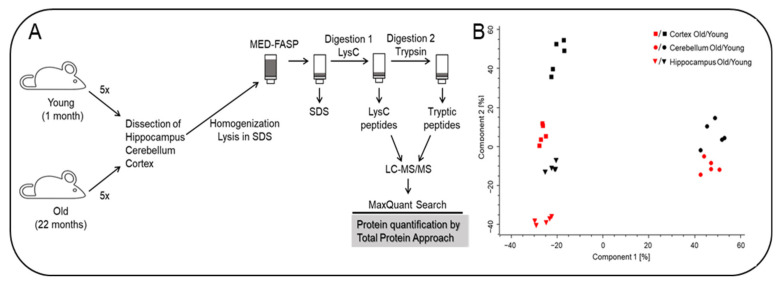
Proteomic analysis of the hippocampus, cortex and cerebellum from young and old mice. (**A**) Proteomic workflow. Brain tissue was isolated from 5 animals per group. The tissue lysates were processed by the MED-FASP procedure with two steps of enzymatic digestion. Peptides were analyzed by LC-MS/MS. The spectra were searched using MaxQuant software. Proteins were quantified by means of Total Protein Approach. (**B**) Principal component analysis of the proteomic data.

**Figure 2 ijms-22-06188-f002:**
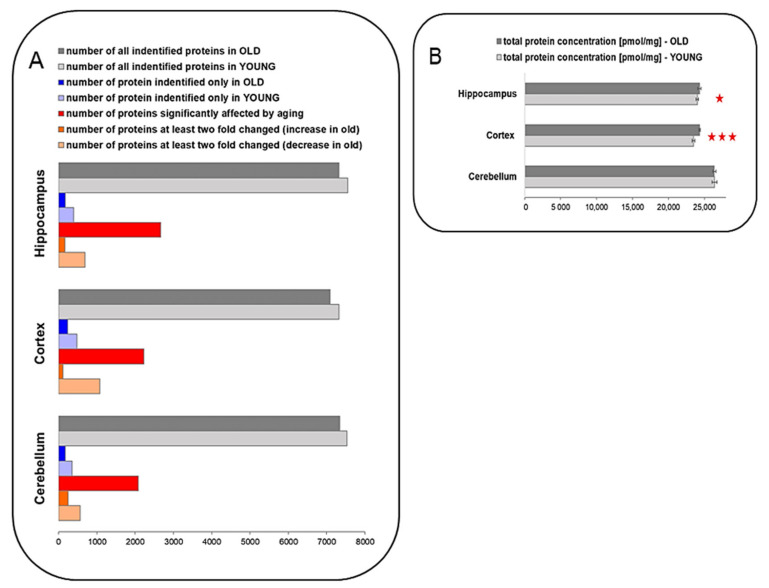
The effect of aging on protein expression in brain structures. (**A**) The comparison of the number of proteins identified in the hippocampus, cortex, and cerebellum. Some of the proteins are unique only for old or young animals. Moreover, approximately 36%, 31%, and 28% of proteins are significantly altered by aging in the hippocampus, cortex, and cerebellum, respectively. We found that 11–16% of the total identified proteins changed at least twofold in old animals. (**B**) Total protein concentration increases significantly in the hippocampus and cortex (* *p* < 0.05 and *** *p* < 0.001, respectively) of the old animals, compared to the young ones. Abbreviations: YOUNG—1-month-old, OLD—22-month-old mice.

**Figure 3 ijms-22-06188-f003:**
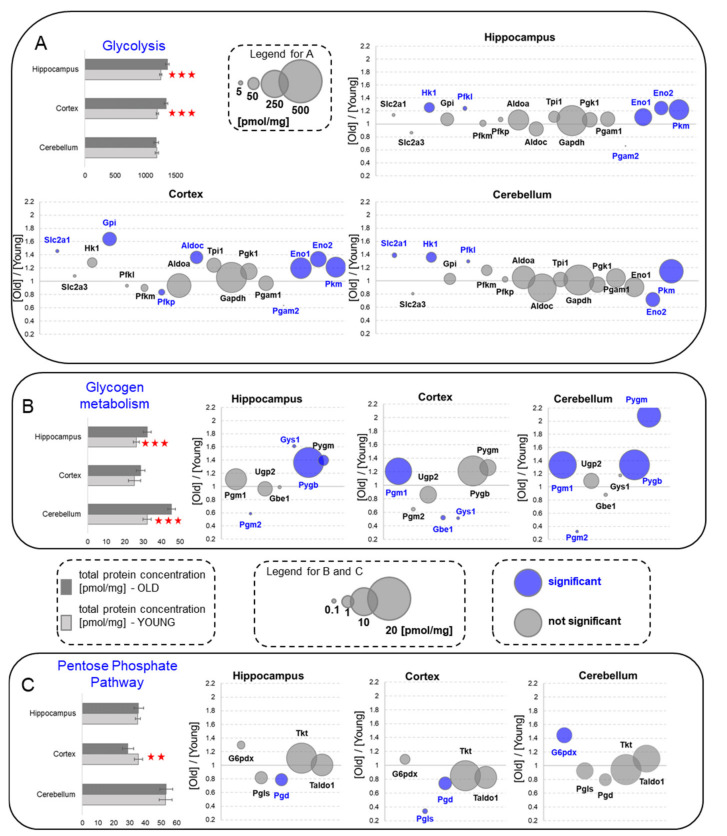
Changes in the energy metabolism processes upon aging—glucose and glycogen as energy sources. (**A**) Glucose transporters and glycolysis. (**B**) Glycogen metabolism. (**C**) Pentose Phosphate Pathway. Column plots show total protein concentration summarizing each metabolic pathway for young and old animals in the hippocampus, cortex, and cerebellum. The statistically significant changes between the age groups are indicated (** *p* < 0.01, *** *p* < 0.001). Bubble plots show ratios of protein concentrations in the old vs. young hippocampus, cortex and cerebellum. The size of the bubbles is proportional to an average protein concentration. Proteins with significantly changed concentrations are highlighted in blue. Abbreviations: YOUNG—1-month-old and OLD—22-month-old mice.

**Figure 4 ijms-22-06188-f004:**
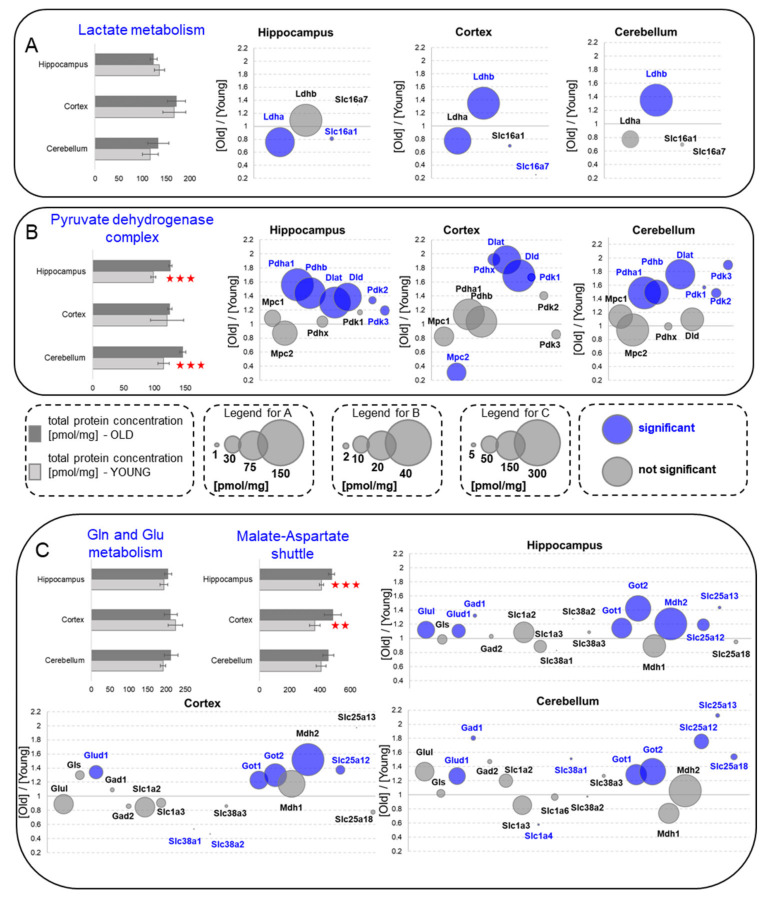
Changes in the energy metabolism processes upon aging—lactate, pyruvate and glutamine as energy sources. (**A**) Lactate metabolism. (**B**) Pyruvate dehydrogenase complex. (**C**) Glutamine/glutamate metabolism with the malate-aspartate shuttle. Column plots show total protein concentration summarizing each metabolic pathway for young and old animals in the hippocampus, cortex and cerebellum. The statistically significant changes between age groups are indicated (** *p* < 0.01, *** *p* < 0.001). Bubble plots show ratios of protein concentrations in the old vs. young hippocampus, cortex and cerebellum. The size of the bubbles is proportional to an average protein concentration. Proteins with significantly changed concentrations are highlighted in blue. Abbreviations: YOUNG—1-month-old and OLD—22-month-old mice.

**Figure 5 ijms-22-06188-f005:**
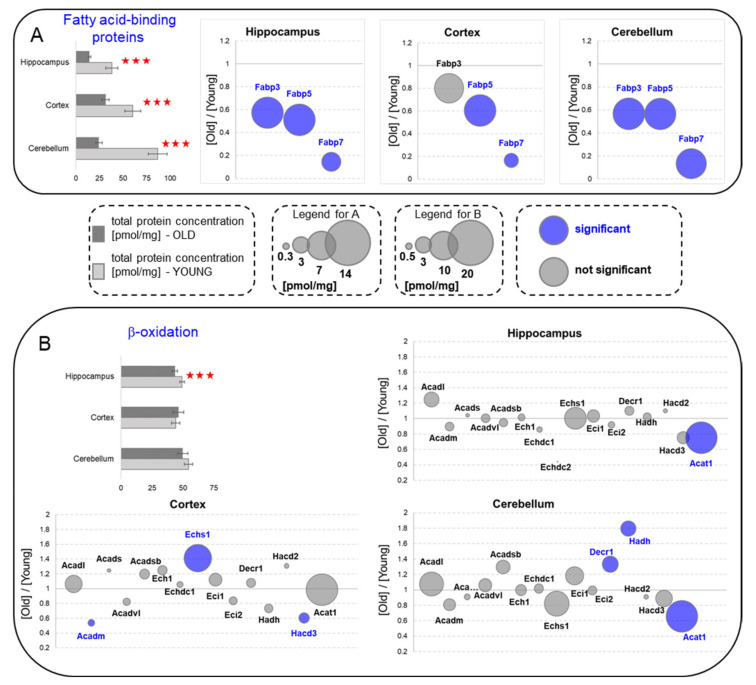
Changes in the energy metabolism processes upon aging—fatty acids as energy sources. (**A**) Fatty acid-binding proteins. (**B**) β-oxidation. Column plots show total protein concentration summarizing each metabolic pathway for young and old animals in the hippocampus, cortex and cerebellum. The statistically significant changes between age groups are indicated (*** *p* < 0.001). Bubble plots show the ratios of protein concentrations in the old vs. the young hippocampus, cortex and cerebellum. The size of the bubbles is proportional to an average protein concentration. Proteins with significantly changed concentrations are highlighted in blue. Abbreviations: YOUNG—1-month-old and OLD—22-month-old mice.

**Figure 6 ijms-22-06188-f006:**
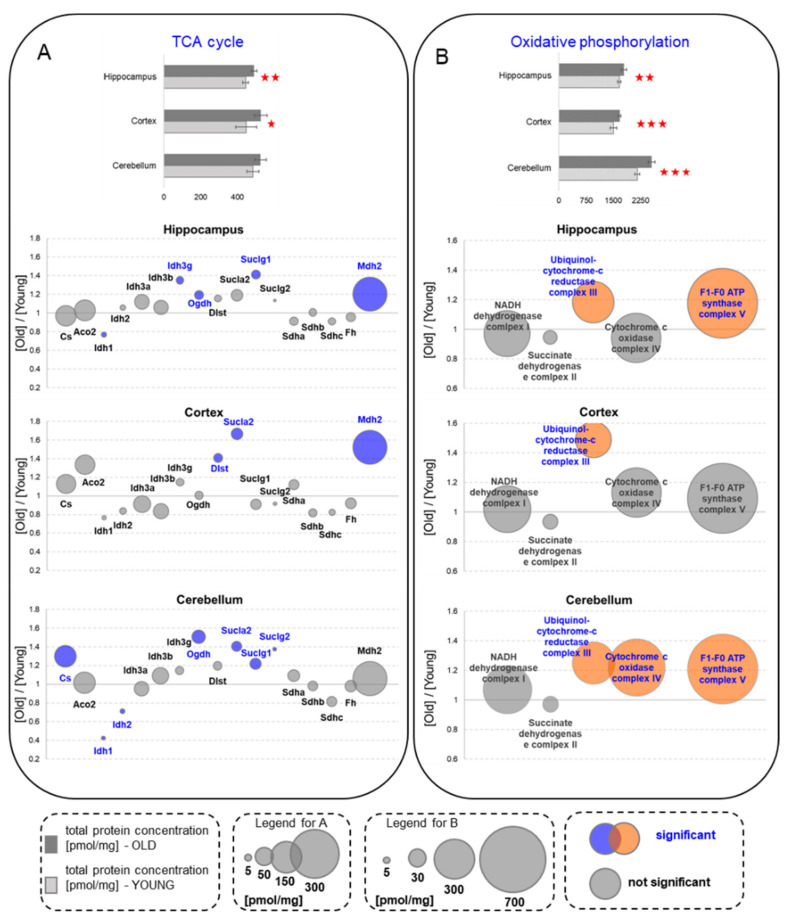
Changes in the energy metabolism processes upon aging—oxidative metabolism. (**A**) Krebs cycle (TCA). (**B**) Oxidative phosphorylation (the plot bubbles refer to the sum of all subunits in each complex). Column plots show total protein concentration summarizing each metabolic pathway for young and old animals in the hippocampus, cortex and cerebellum. The statistically significant changes between age groups are indicated (* *p* < 0.05; ** *p* < 0.01, *** *p* < 0.001). Bubble plots show ratios of protein concentrations in the old vs. young hippocampus, cortex and cerebellum. The size of the bubbles is proportional to an average protein concentration. Proteins with significantly changed concentrations are highlighted in blue in A and orange in the B panel. Abbreviations: YOUNG—1-month-old and OLD—22-month-old mice.

**Figure 7 ijms-22-06188-f007:**
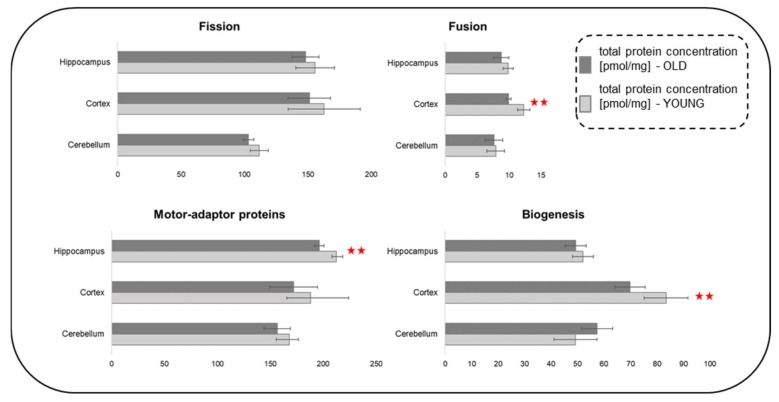
Proteins regulating mitochondrial network dynamics—fusion fission, motility and biogenesis. The statistically significant changes between age groups are indicated (** *p* < 0.01). Abbreviations: YOUNG—1-month-old and OLD—22-month-old mice.

## Data Availability

The data presented in this study are available on request from the corresponding authors.

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
