# Peer review of "Absolute Proteome Analysis of Hippocampus, Cortex and Cerebellum in Aged and Young Mice Reveals Changes in Energy Metabolism"

_ijms, 2021, doi:10.3390/ijms22126188_

Round 1
Reviewer 1 Report
The authors performed proteome analysis in three different regions of brain in young and aged mice. They demonstrated that aging did not affect significantly the abundance of total proteins in the brain structures, but the levels of proteins constituting energy metabolism pathways differed between young and aged mice. This is an interesting and extensive study and acceptable for publication. However, this reviewer has some concerns.
- It is not clear why they authors focused on the three mentioned brain regions (cortex, hippocampus and cerebellum). Please make it clear in Introduction.
- The authors studied only male mice. However there is also possibility of sex differences in energy metabolism pathways. I would suggest to add female mice in their study.
- Please separate the results and Discussion chapter for better understanding.
- Please add the law number/code number for local Ethical commission.
- Please add the bregma of brain region in Materials and Method (preparation of brain analysis), which will give us a clear idea about the region studied. The protein concentrations are varied sometimes for dorsal and ventral hippocampus. What parts of hippocampus is studied is not clear for the readers.
Author Response
Thank you for the review and helpful comments. Please find our detailed response below.
- It is not clear why they authors focused on the three mentioned brain regions (cortex, hippocampus and cerebellum). Please make it clear in Introduction.
Response: We shortly explained (in the Introduction section) why these three brain formations were studied:
Lines 58-61 “The role aging-related physiological changes in cerebral cortex and hippocampus in cognitive performance is undisputed. However, it has been shown that also changes within cerebellum result in cognitive decline [e.g., see 10].
We added the citation to the reference list and changed the numbering accordingly.
- The authors studied only male mice. However there is also possibility of sex differences in energy metabolism pathways. I would suggest to add female mice in their study.
Response: We agree with the Reviewer that some sex differences may exist and we are trying to raise the old mice of the opposite sex to compare the results with these presented in this manuscript.
We are especially grateful to the Reviewer for this comment as in fact, we studied the female mice, and we somehow lost the prefix “fe” in the sentence of Materials and Methods (3.1. Animals and tissue preparation). We corrected this.
- Please separate the results and Discussion chapter for better understanding.
Response: We considered this possibility while writing this paper, but came to the conclusion that in this case, the Results section without commentaries would be very dry, hard to read and absorb, and not too informative, especially for readers not familiar with energy metabolism of the brain.
However, we thought that perhaps in such a situation, a more exhausted chapter “Conclusions” would be beneficial. Thus, in the corrected version of the manuscript, the chapter is expanded. We hope that the Reviewer finds it satisfactory.
- Please add the law number/code number for local Ethical commission.
Response: We added this information to the subchapter “3.1. Animals and tissue preparation”
- Please add the bregma of brain region in Materials and Method (preparation of brain analysis), which will give us a clear idea about the region studied. The protein concentrations are varied sometimes for dorsal and ventral hippocampus. What parts of hippocampus is studied is not clear for the readers.
Response: We agree with the Reviewer that expression of proteins may differ among various parts of hippocampus (as well as cortex and cerebellum). In this manuscript, however, we studied the titers of proteins in the whole right half of brain’s formations: hippocampus (the whole right hippocampus), cerebellum (right cerebellar hemisphere) and cortex (the whole right isocortex) and information about it we added to the subchapter “3.2. Preparation of tissue lysates”:
“Immediately after isolation the brain structures from the right half of brain’s: hippocampus (the whole right hippocampus), cerebellum (right cerebellar hemisphere) and cortex (the whole right isocortex)”
Reviewer 2 Report
In the submitted manuscript the authors use massspectrometry to define the proteomic in different brain areas (cerebellum, cortex and hippocampus) in young and aged mice, with a focus on metabolism and energy homeostasis. The manuscript is overall well structured and reported data are significant.
My only query is as follow:
Furthermore, in spite lane 68, Figure 1A does not provide information on cerebellum, cortex and hippocampus dissection: what about the brain cut sites?
Author Response
Thank you for the review and helpful comments. Please find our detailed response below.
"My only query is as follow: Furthermore, in spite lane 68, Figure 1A does not provide information on cerebellum, cortex and hippocampus dissection: what about the brain cut sites?"
Response:
In the corrected version of the manuscript we added more precise information about the studied structures (“3.2. Preparation of tissue lysates”: “Immediately after isolation the brain structures from the right half of brain’s: hippocampus (the whole right hippocampus), cerebellum (right cerebellar hemisphere) and cortex (the whole right isocortex)”.
Moreover, we changed the order of affiliations, because the first author has double affiliation and the measurements presented in this manuscript have been done in the Max-Planck Institute.
We also corrected the number of the grant (section: „Funding”) because we have noticed that we pasted a wrong one.
Additionally, according to the Editor’s comments, we made some changes in the Materials and Methods section.
Round 2
Reviewer 1 Report
The manuscript is now improved and can be accepted in present form.